# The plastome sequence of *Bactris gasipaes* and evolutionary analysis in tribe Cocoseae (Arecaceae)

Raquel Santos da Silva[1], Charles Roland Clement[2], Eduardo Balsanelli[3,4], Valter Antonio de Baura[3], Emanuel Maltempi de Souza[3], Hugo Pacheco de Freitas Fraga[1], Leila do Nascimento Vieira[1]*

1 Departamento de Botânica, Universidade Federal do Paraná, Curitiba, Paraná, Brazil, 2 Coordenação de Tecnologia e Inovação, Instituto Nacional de Pesquisas da Amazônia, Manaus, AM, Brazil, 3 Departamento de Bioquímica e Biologia Molecular, Núcleo de Fixação Biológica de Nitrogênio, Universidade Federal do Paraná, Curitiba, Paraná, Brazil, 4 Departamento de Bioquímica e Biologia Molecular, GoGenetic, Universidade Federal do Paraná, Curitiba, Paraná, Brazil

* leilanvieira@gmail.com

**Data Availability Statement:** The plastome sequence was deposited in GenBank: MW054718.

## Abstract

The family Arecaceae is distributed throughout tropical and subtropical regions of the world. Among the five subfamilies, Arecoideae is the most species-rich and still contains some ambiguous inter-generic relationships, such as those within subtribes Attaleinae and Bactridineae. The hypervariable regions of plastid genomes (plastomes) are interesting tools to clarify unresolved phylogenetic relationships. We sequenced and characterized the plastome of *Bactris gasipaes* (Bactridinae) and compared it with eight species from the three Cocoseae sub-tribes (Attaleinae, Bactridinae, and Elaeidinae) to perform comparative analysis and to identify hypervariable regions. The *Bactris gasipaes* plastome has 156,646 bp, with 113 unique genes. Among them, four genes have an alternative start codon (*cemA*, *rps19*, *rpl2*, and *ndhD*). Plastomes are highly conserved within tribe Cocoseae: 97.3% identity, length variation of ~2 kb, and a single ~4.5 kb inversion in *Astrocaryum* plastomes. The LSC/IR and IR/SSC junctions vary among the subtribes: in Bactridinae and Elaeidinae the *rps19* gene is completely contained in the IR region; in the subtribe Attaleinae the *rps19* gene is only partially contained in the IRs. The hypervariable regions selected according to sequence variation (SV%) and frequency of parsimony informative sites (PIS%) revealed plastome regions with great potential for molecular analysis. The ten regions with greatest SV% showed higher variation than the plastid molecular markers commonly used for phylogenetic analysis in palms. The phylogenetic trees based on the plastomes and the hypervariable regions (SV%) datasets had well-resolved relationships, with consistent topologies within tribe Cocoseae, and confirm the monophyly of the subtribes Bactridinae and Attaleinae.

## Introduction

The family Arecaceae contains 181 genera and about 2,600 species distributed throughout tropical and subtropical regions of the world [1]. The most recent taxonomic review in

**Funding:** This research was supported by Conselho Nacional de Desenvolvimento Científico e Tecnológico (CNPq - https://www.gov.br/cnpq) [grant numbers LNV 435200/2018-6, CRC 303477/2018-0]. This research was also partially supported by The Brazilian Program of National Institutes of Science and Technology-INCT/ Brazilian Research Council-CNPq/MCT and Embrapa's Portfólio de Recursos Genéticos (https://www.embrapa.br/), which helped to maintain the Peach palm Active Germplasm Bank at INPA. The funders had no role in study design, data collection and analysis, decision to publish, or preparation of the manuscript.

**Competing interests:** The authors have declared that no competing interests exist.

Arecaceae, published by Baker and Dransfield [1], recognizes five subfamilies: Arecoideae, Calamoideae, Ceroxyloideae, Coryphoideae, and Nypoideae. Arecoideae is the largest subfamily, with 14 tribes and 108 genera. Several Arecoideae tribes have been extensively studied, but some inter-generic relationships remain ambiguous, such as those within subtribes Attaleinae and Bactridineae [1]. These subtribes are both within tribe Cocoseae, which includes Elaeidinae as its third subtribe.

Plastid genomes (plastomes) are a useful tool for phylogenetic and evolutionary studies [2]. Hypervariable regions in plastomes can provide information to elucidate phylogenetic relationships that are not yet well resolved [3, 4]. However, these highly variable regions vary between clades, and their identification may be necessary for each taxonomic level [3]. Angiosperms show structural rearrangements, loss of genes, introns, and heterogeneous nucleotide substitution rates in protein-coding genes among their plastomes [5, 6]. Also, the fact that palms have a low mutation rate in the plastome [7] makes the identification of clade-specific hypervariable regions singularly relevant for the group.

In addition to use in phylogenetic analysis, these hypervariable regions may also be powerful molecular markers for analysis of population genetic structure, including that among wild and domesticated populations. The family Arecaceae includes species of great ecological importance, either by their interaction with pollinators [8] or with frugivorous animals [9, 10]. Also, palms provide several high-value products for industry (e.g., fibers, construction materials, oil, medicinal compounds, heart-of-palm, fruits) and are especially important for tropical and subtropical indigenous and traditional communities [11, 12]. Among the most useful palms, the coconut (*Cocos nucifera* L.), the date palm (*Phoenix dactylifera* L.), and the oil palm (*Elaeis guineensis* Jacq.) stand out by their global economic importance. The peach palm (*Bactris gasipaes* Kunth) is the only fully domesticated palm in the Neotropics [11]; it was domesticated for its fruit and is becoming important as a source of heart-of-palm [13]. Many Cocoseae species included in our analysis have interesting domestication histories, such as *Bactris gasipaes*, *Elaeis guineensis*, *Cocos nucifera* [11, 14, 15]. Others are from genera with traditional use by indigenous and traditional communities in South America, such as *Butia*, *Astrocaryum*, *Acrocomia*, and *Syagrus* [16–19].

We sequenced and characterized the plastome of *Bactris gasipaes* and compared it with eight species from the three sub-tribes (Attaleinae, Bactridinae, and Elaeidinae) of the tribe Cocoseae to perform comparative structural analysis and phylogenetic inference, and to identify hypervariable regions.

## Material and methods

### Taxon sampling

We collected fresh leaves from a wild individual of *Bactris gasipaes*, i.e., var. *chichagui* type 1 ([20]; accession number F0205/83) in the core collection of Peach palm Active Germplasm Bank [21] at the National Research Institute for Amazonia (INPA, Manaus, AM, Brazil).

Plastomes and nuclear marker sequences from seven genera of tribe Cocoseae were downloaded from GenBank, including individuals from its three subtribes: Attaleinae (3 genera; 3 species), Elaeidinae (1 species), and Bactridinae (3 genera; 4 species). The species names and the GenBank accession numbers for sequences used in the analysis are in S1 Table.

### *Bactris gasipaes* plastome sequencing

The extraction of plastid-enriched DNA was based on the methodology described by Vieira et al. [22] and modified by Sakaguchi et al. [23], proportionally adjusting the buffer volumes for 8 g of fresh leaves. The DNA extraction was performed with CTAB buffer, as described by

Shi et al. [24]. The DNA was purified with the Genomic DNA Clean & Concentrator™-10 Kit (Zymo Research, Irvine, CA, USA). The purified DNA was quantified using Qubit™ dsDNA HS Assay kit (Thermo Fisher Scientific, Carlsbad, CA, USA) in Qubit™ Fluorometer (Thermo Fisher Scientific). Libraries were prepared with Nextera XT DNA Library Preparation Kit (Illumina, San Diego, CA, USA) and sequenced in Illumina MiSeq® (Illumina), obtaining 250 bp paired-end reads.

Plastome assembly was performed using CLC Genomics Workbench v.8.0 (Qiagen, Germantown, MD, USA) software with *de novo* strategy. The *Acrocomia aculeata* plastome was used as a reference for the ordering of contigs. Plastome annotation was performed using Geneious Prime® (Biomatters, Auckland, New Zealand). For all genes, manual verification was performed, adjusting the initial and terminal codons. The final plastome sequence was deposited in GenBank: MW054718.

## Plastome structural analysis in tribe Cocoseae

The comparative analysis to identify structural rearrangements in the plastomes of the Cocoseae species was carried out using eight species (S1 Table), excluding one IR from all plastomes, and using the progressive alignment on Mauve software [25]. The IRScope software [26] was used to visualize and compare the plastome junctions (IRb/LSC; IRb/SSC, SSC/IRa; IRa/LSC).

## Identification of hypervariable regions

We estimated the variability of the sequences with the formula proposed by Shaw et al. [27], adapted and used by Zavala-Páez et al. [28]. First, we individually aligned each collinear coding sequence (CDS), intergenic spacers (IGS), and introns of the plastomes (list of species in S1 Table) using MAFFT v.7 software [29]. Then, the alignments were imported into the software DNAsp v6.12.03 [30] to obtain the number of invariable sites (monomorphic), parsimony informative sites (PIS), number of substitutions, and number of InDel events. Sequence variability (SV) was calculated using the formula: *SV% = [(number of substitutions + number of InDels) / (number of substitutions + number of InDels + invariable sites)] x 100*. The frequency of PIS was calculated using the formula: *PIS% = [(number of parsimony informative sites/ number of substitutions + number of InDels + invariable sites)] x 100*.

The ten regions with the highest SV% and PIS% values were selected to carry out the subsequent analysis. The plastid markers *matK*, *trnQ-rps16*, *rps16* intron, *trnD-trnT*, *trnL-trnF* [31, 32] and the nuclear markers PRK and RPB2 [32, 33], commonly used for phylogenetic analysis in Arecaceae, were used for comparative purposes and subjected to the same procedure to obtain the PIS% and SV% values.

## Phylogenetic inferences

Phylogenetic inferences were made including the following species of the tribe Cocoseae: *Bactris gasipaes*, *Acrocomia aculeata*, *Astrocaryum aculeatum*, *Astrocarym murumuru*, *Butia eriospatha*, *Cocos nucifera*, *Elaeis guineensis*, *Syagrus coronata*, and two species as outgroup: *Brahea brandegeei* (Purpus) H. E. Moore (subfamily Coryphoideae) and *Archontophoenix alexandrae* (F.Muell.) H.Wendl. & Drude (subfamily Arecoideae). Three data sets were used: i) the plastome alignment (one IR excluded); ii) the ten regions with the greatest SV% value; iii) the ten regions with the greatest PIS% value.

Plastome alignment was performed using progressive alignment on Mauve [25] implemented in Geneious Prime® v.2020.1.2. The Locally Collinear Blocks (LCBs) identified by Mauve were individually extracted and concatenated. The alignment of the ten regions with

**Table 1. Substitution models selected for the phylogenetic inferences using Maximum Likelihood (ML).**

| Region | Models for ML |
|---|---|
| Plastome | K3Pu+F+R2 |
| *trnC-petN* | HKY+F |
| *psbC-trnS* | JC+I |
| *psaC-ndhE* | F81+F+G4 |
| *ccsA-ndhD* | F81+F+I |
| *petN-psbM* | TPM2+F+I |
| *accD-psaI* | F81+F+I |
| *trnS-trnG* | F81+F+I |
| *rps15-ycf1* | HKY+F+I |
| *ndhF-rpl32* | F81+F+I |
| *rpl16-intron* | F81+F+I |
| *petD-rpoA* | F81+F+I |
| *petA-psbJ* | F81+F+I |
| *trnG-trnfM* | F81+F+I |
| *rps8-rpl14* | F81+F+I |

the highest SV% and PIS% values was carried out using MAFFT v.7.450 [29] implemented in Geneious Prime® v.2020.1.2. Phylogenetic inferences were performed by Maximum Likelihood (ML) using W-IQ-tree [34], with 1,000 bootstrap repetitions. The choice of substitution models, including FreeRate heterogeneity model, was made according to Bayesian information criterion (BIC; Table 1). Branch support analysis was performed with 1,000 repetitions of bootstrap and single branch test SH (-aLTR, 1,000 replicates). The resulting trees were represented using Geneious Prime® v.2020.1.2.

## Results

### *Bactris gasipaes* plastome

The sequencing of plastid-enriched DNA resulted in 448,600 reads with an average length of 214 bp. Of these, 47,735 were plastome reads (~10%), resulting in an average depth of coverage of 67.64 (SD = 24.32). The assembled plastome has a 21 bp gap in the IGS *trnT-UGU/ trnL-UAA* (position 46,800 to 46,820). This gap is in an AT-rich region (sequence 22 bp upstream to 119 bp downstream is only 7.8% of GC-content) and is, therefore, difficult to sequence [35]. We calculated this gap length using other species of tribe Cocoseae as reference.

*Bactris gasipaes* plastome has the quadripartite structure typically found in angiosperms [2], with a pair of inverted repeat (IRs), a large single-copy region (LSC), and a small single-copy region (SSC). The IRs are 27,038 bp in length (each), the LSC is 85,118 bp, and the SSC is 17,452 bp, resulting in a plastome with 156,646 bp.

*Bactris gasipaes* plastome has an average GC-content of 37.5%. When comparing the plastome regions, the SSC has the lowest GC-content, with 31.3%, followed by LSC with 35.5%. The IRs have the highest value, with 42.6% of GC-content. The rRNA and tRNA show high GC-content, with 55.3% and 53.4%, respectively. Protein-coding genes have an average GC-content of 37.9%, intergenic spacers (IGS) of 37.5%, and introns of 37.1%. The plastome GC-content among species from tribe Cocoseae is similar, ranging from 37.40% (*Elaeis guineensis*) to 37.53% (*Acrocomia aculeata*).

In the *Bactris gasipaes* plastome, we annotated 113 unique genes, 79 of which are protein-coding genes, 30 tRNA genes, and 4 rRNA genes (Table 2). Duplicate genes in IRs include 8

**Table 2. List of genes of *Bactris gasipaes* plastome organized according to their location.**

| Plastome region | Name of genes |
|---|---|
| Large Single Copy (LSC) | *psbA, trnK-UUU\*, matK, rps16\*, trnQ-UUG, psbK, psbI, trnS-GCU, trnG-UCC\*, trnR-UCU, atpA, atpF\*, atpH, atpI, rps2, rpoC2, rpoC1\*, rpoB, trnC-GCA, petN, psbM, trnD-GUC, trnY-GUA, trnE-UUC, trnT-GGU, psbD, psbC, trnS-UGA, psbZ, trnG-GCC, trnfM-CAU, rps14, psaB, psaA, ycf3\*, trnS-GGA, rps4, trnT-UGU, trnL-UAA\*, trnF-GAA, ndhJ, ndhK, ndhC, trnV-UAC\*, trnM-CAU, atpE, atpB, rbcL, accD, psaI, ycf4, cemA, petA, psbJ, psbL, psbF, psbE, petL, petG, trnW-CCA, trnP-UGG, psaJ, rpl33, rps18, rpl20, rps12\* (exon 1), clpP\*, psbB, psbT, psbN, psbH, petB\*, petD\*, rpoA, rps11, rpl36, infA, rps8, rpl14, rpl16\*, rps3, rpl22* |
| Inverted Repeat (IR) | *rps19, trnH-GUG, rpl2\*, rpl23, trnI-CAU, ycf2, trnL-CAA, ndhB\*, rps7,* rps12\* (exons 2 and 3)*, trnV-GAC, rrn16, trnI-GAU\*, trnA-UGC\*, rrn23, rrn4.5, rrn5, trnR-ACG, trnN-GUU* |
| IR / SSC junction | *ndhF, ycf1* |
| Small Single Copy (SSC) | *rpl32, trnL-UAG, ccsA, ndhD, psaC, ndhE, ndhG, ndhI, ndhA\*, ndhH, rps15* |

\* Intron-containing genes.

tRNA genes, 4 rRNA genes, and 7 protein-coding genes (Table 2). Among the 113 genes, 15 genes contain one intron (6 tRNA genes and 9 protein-coding genes) and 3 genes contain two introns (*clpP, ycf3*, and *rps12*; Table 2). Among intron-containing genes, 12 are located in LSC (*trnK-UUU, rps16, trnG-UCC, atpF, rpoC1, ycf3, trnL-UAA, trnV-UAC, clpP, petB, petD, rpl16*), 1 in SSC (*ndhA*), 4 in IRs (*rpl2, ndhB, trnI-GAU, trnA-UGC*), and *rps12* is a trans-splicing gene with the first exon located in the LSC region and the second and third exons in the IRs.

Surprisingly, the *cemA* gene exhibited an alternative start codon, as reported in species of subtribes Attaleinae and Elaeidinae, contrasting with the Bactridinae species sequenced so far (*Astrocaryum aculeatum, Astrocaryum murumuru*, and *Acrocomia aculeata*). Three other genes have alternative initiation codons, *rps19* (GTG), *rpl2* (ACG), *ndhD* (ATC).

## Comparative analysis in tribe Cocoseae

Plastomes are highly conserved (97.3% identity) within tribe Cocoseae. Species from subtribes Bactridinae and Elaeidinae have a plastome ~2 kb larger than the species from subtribe Attaleinae. This difference in length between the subtribes is mainly in the IRs and LSC regions (Table 3). The plastomes from Cocoseae species ranged from 154.048 bp (*Butia eriospatha*) to 156.937 bp (*Elaeis guineenses*).

The progressive alignment among species from tribe Cocoseae shows evidence for three Locally Collinear Blocks (LCBs) (Fig 1). These three LCBs are a result of the 4.5 kb inversion

**Table 3. Plastomes of tribe Cocoseae.**

| Subtribe | Species | Plastome (bp) | LSC (bp) | IR (bp) | SSC (bp) |
|---|---|---|---|---|---|
| Elaeidinae | *Elaeis guineensis* | 156,973 | 85,192 | 27,071 | 17,639 |
| Bactridinae | *Astrocaryum aculeatum* | 156,804 | 85,037 | 27,081 | 17,605 |
| | *Astrocaryum murumuru* | 156,801 | 85,017 | 27,081 | 17,622 |
| | *Bactris gasipaes* | 156,646 | 85,118 | 27,038 | 17,452 |
| | *Acrocomia aculeata* | 156,500 | 84,936 | 27,092 | 17,380 |
| Attaleinae | *Syagrus coronata* | 155,053 | 84,535 | 26,522 | 17,474 |
| | *Cocos nucifera* | 154,731 | 84,230 | 26,555 | 17,391 |
| | *Butia eriospatha* | 154,048 | 83,805 | 26,437 | 17,369 |

LSC: large single copy region; IR: inverted repeat; SSC: small single copy region.

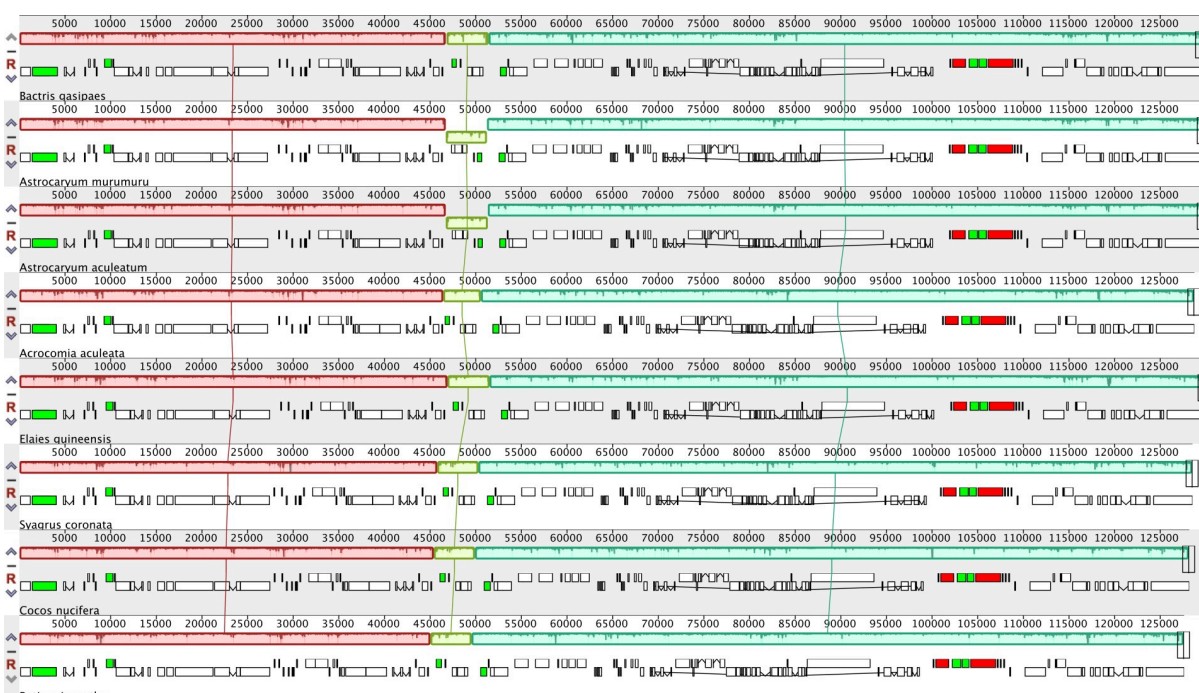

**Fig 1. Plastome rearrangement analysis within tribe Cocoseae.** Locally collinear blocks (LCBs) are identified by colors. The 4.5 kb inversion in *Astrocaryum* is in green.

present in the plastomes of *Astrocaryum murumuru* and *Astrocaryum aculeatum* (Fig 1). The set of genes that makes up this structural rearrangement is composed of *ndhC*, *ndhK*, *ndhJ*, *trnF-GAA*, and *trnL-UAA*.

In the LSC/IR and IR/SSC junctions of the plastomes, there are differences among the subtribes (Fig 2). In Bactridinae (*Acrocomia aculeata*, *Astrocaryum aculeatum*, *Astrocaryum murumuru*, and *Bactris gasipaes*) and Elaeidinae (*Elaeis guineensis*) the *rps19* gene is completely contained in the IR region and, therefore, there are two copies of the complete gene. In contrast, in the subtribe Attaleinae (*Butia eriospatha*, *Cocos nucifera*, and *Syagrus coronata*) the *rps19* gene is only partially contained in the IRs, resulting in a complete *rps19* and a partial *rps19*: the complete *rps19* gene starts at IRb and ends at LSC (LSC/IRb); the partial *rps19* starts at IR, but does not contain the final portion of the gene (Fig 2).

Similarly, the *ycf1* gene is partially contained in IRs, with a complete *ycf1* at IRa/SSC and a partial (pseudo) *ycf1* at IRb. The *ndhF* gene has both position and length conserved at the IRb/SSC junction in tribe Cocoseae, with the portion of the gene contained in the IRb overlapping the *ycf1* gene (56 bp) (Fig 2).

## Hypervariable regions

We carried out the SV% and PIS% estimates to identify the plastome regions with the greatest variation within tribe Cocoseae. All ten regions selected according to the highest SV% showed greater variation than the plastid molecular markers commonly used for phylogenetic analysis in palms (Fig 3A). As expected, they have SV% lower than the nuclear markers PRK and RPB2 (Fig 3A). Among the ten regions selected according to the highest PIS% values, all showed greater values than the plastid molecular markers commonly used for phylogenetic analysis in palms (Fig 3B) and two of them (*trnC-petN* and *psbC-trnS*) were more variable than the

# Inverted Repeats

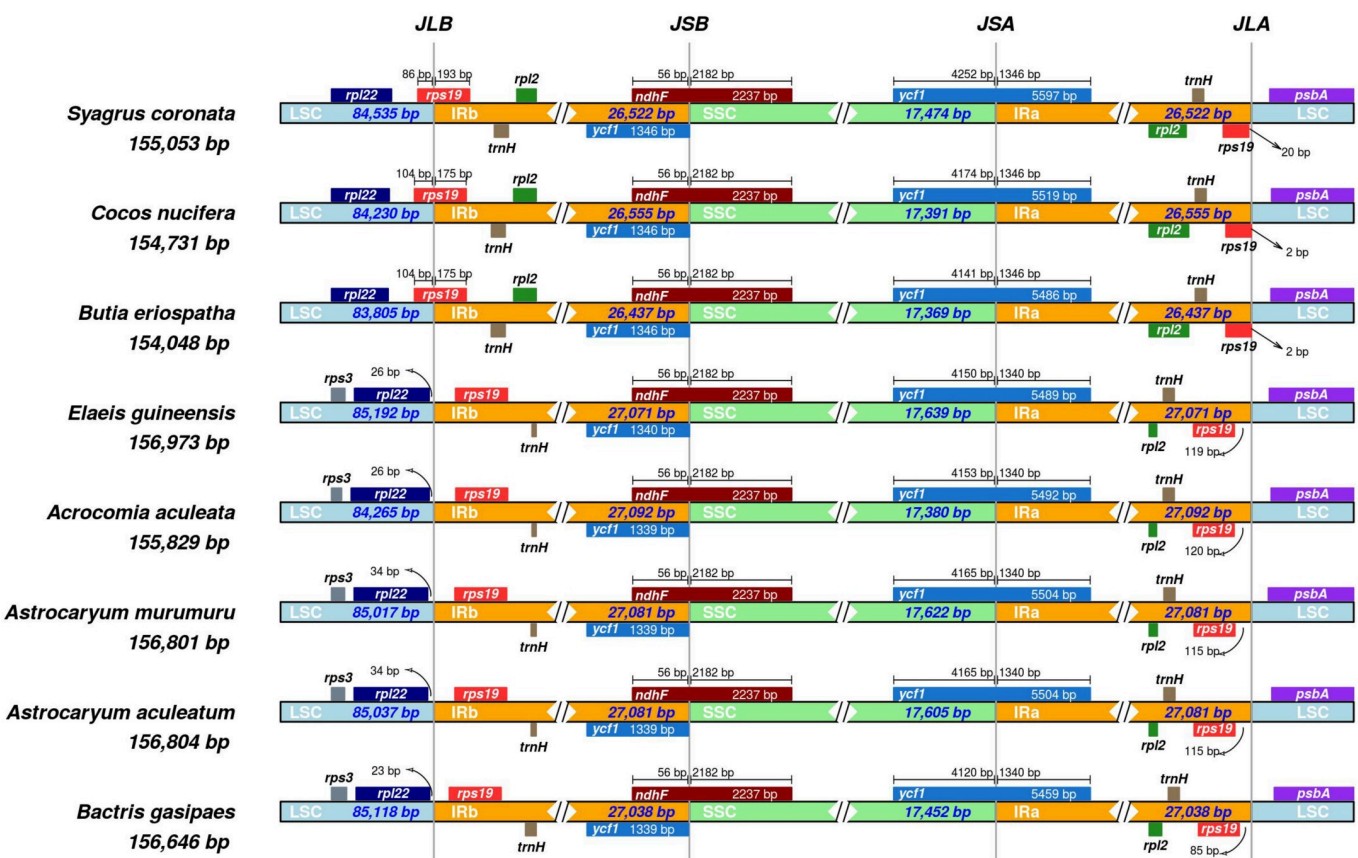

**Fig 2. Comparison of plastome junctions (IRb/LSC; IRb/SSC, SSC/IRa; IRa/LSC) among Cocoseae species.** The numbers indicate sequence length in base pairs.

nuclear marker PRK (Fig 3B). The nuclear marker RPB2 showed the highest values for both SV% and PIS% estimates.

We also calculated the frequency of substitutions and frequency of InDel events for each plastome region. The substitution frequency was ~5x higher than InDels in plastomes (Table 4). In the coding sequences (CDSs), substitutions are ~80x more common than InDels. In IGSs and introns, substitutions are ~4x and ~3x more frequent than InDels, respectively (Table 4). In general, these data show that in all regions of Cocoseae plastomes there is a higher frequency of substitutions than InDels.

## Phylogenomic analysis

The complete LCB matrix of the aligned plastomes consisted of 137,452 columns, of which 134,244 are constant and 815 parsimony-informative. The SV matrix contained a total of 9614 columns, with 133 parsimony-informative and 9144 constant sites, and the PIS matrix 7538 columns, with 129 parsimony-informative and 7200 constant sites. The phylogenomic trees inferred using Maximum Likelihood (ML) and based on the plastome and the ten selected regions with greatest SV% datasets showed identical topologies (Fig 4) and high bootstrap

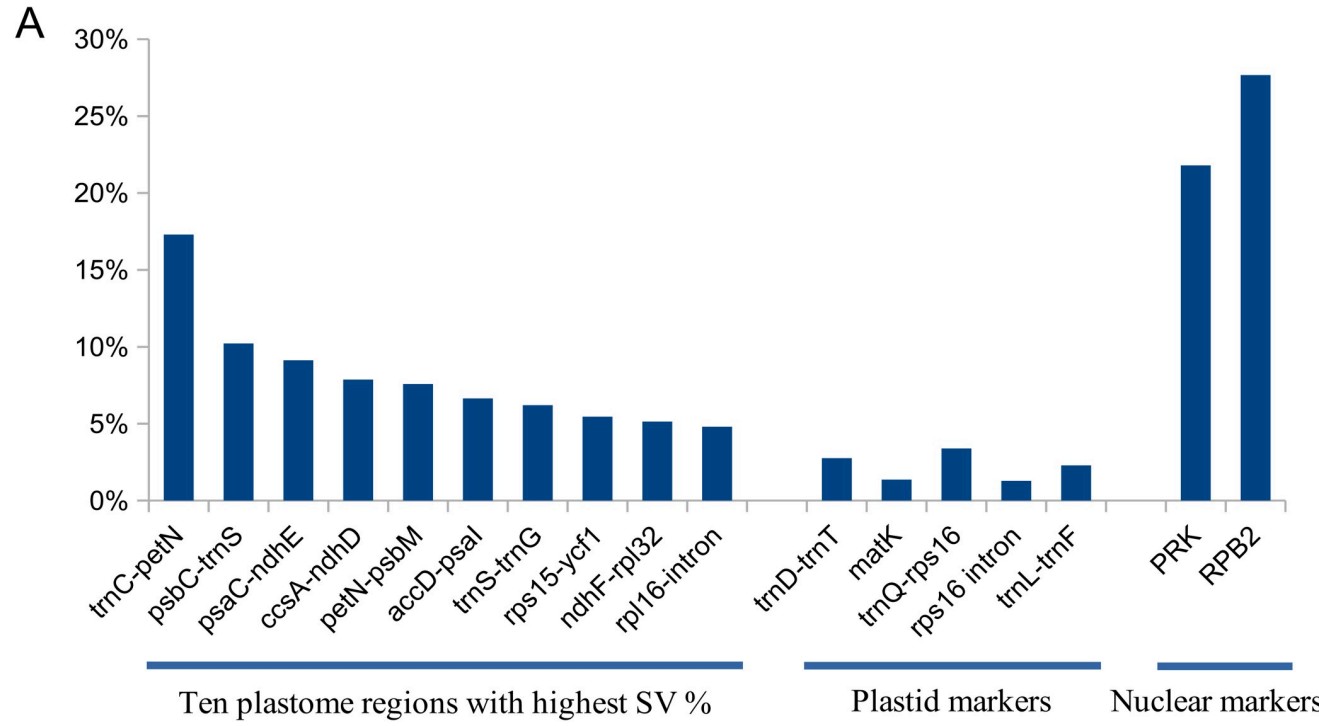

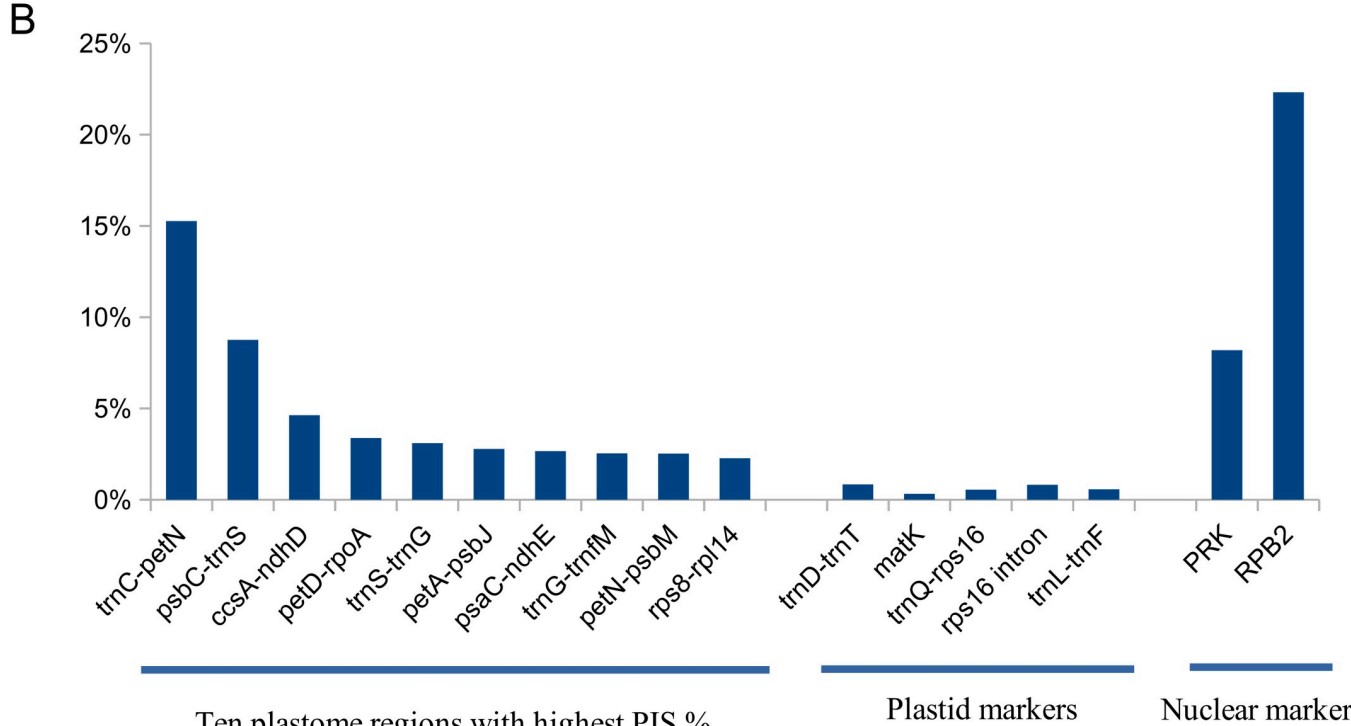

**Fig 3. Hypervariable plastome regions compared with commonly used plastid and nuclear markers. A**—The ten plastome regions with greatest sequence variation (SV%); **B**—The ten plastome regions with greatest frequency of parsimony informative sites (PIS%).

**Table 4. Frequency of substitutions/mutations and insertions/deletions (InDels) among the plastome sequence.**

|  | Plastome | CDS | IGS | Introns |
|---|---|---|---|---|
| Mutations (%) | 1.53 | 0.80 | 2.30 | 1.12 |
| InDels (%) | 0.30 | 0.01 | 0.58 | 0.34 |
| Mutation / InDel ratio | 5.02 | 79.90 | 3.99 | 3.26 |

CDS–coding sequence; IGS–intergenic spacer.

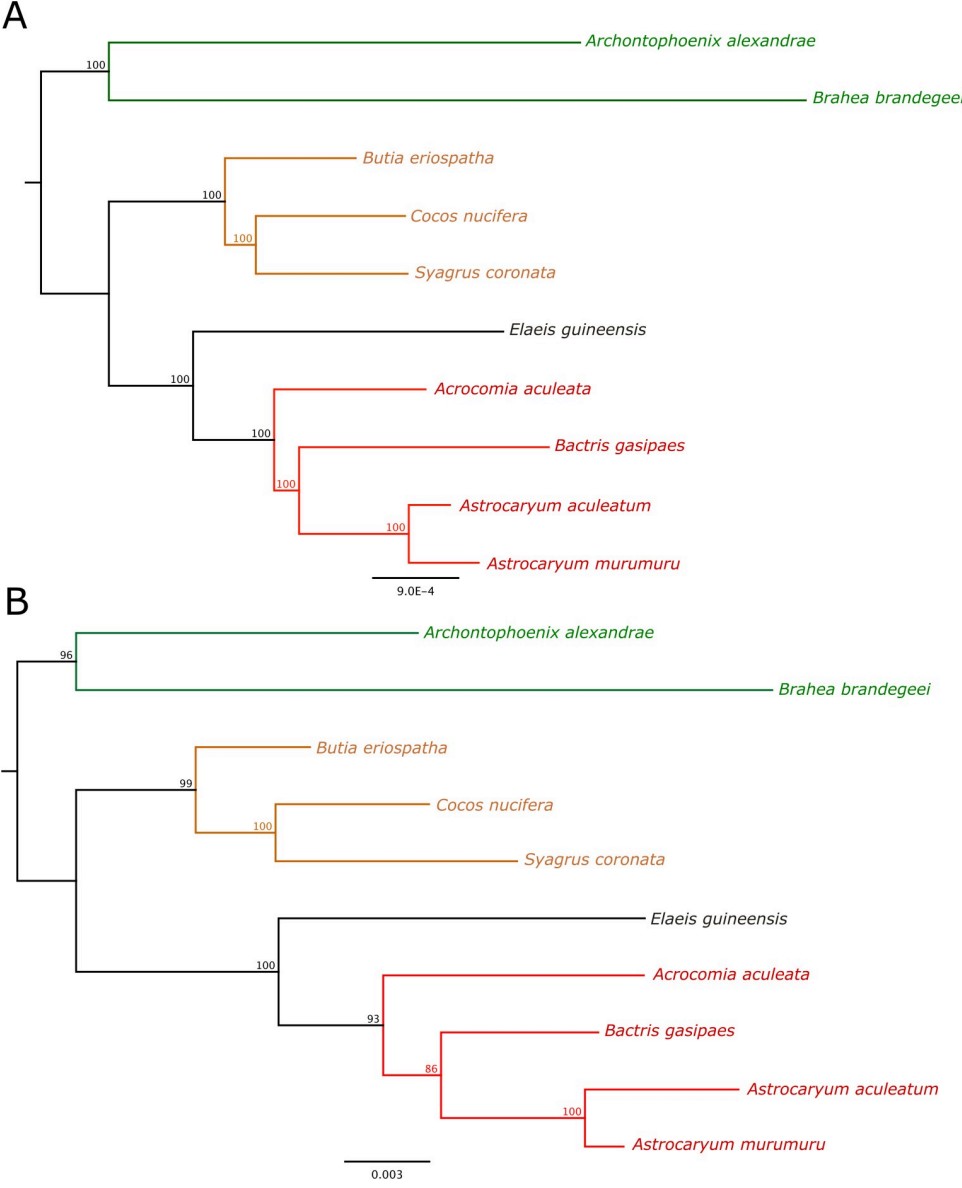

**Fig 4. Phylogenetic trees based on maximum likelihood inference. A**–Phylogenetic inference using plastome sequences (one IR removed); **B**—Phylogenetic inference using the ten plastome regions with greatest sequence variability (SV%) values. The numbers above the branches are maximum likelihood bootstrap values (1000 replicates).

support (> 86). The phylogenetic tree inferred from the alignment of the ten regions with greatest PIS values showed a similar topology; it differed only by the presence of a polytomy in the Attaleinae clade, generated by the low bootstrap support in intergeneric relationships (S1 Fig). For the topologies generated by ML in the three datasets, the monophyly of the subtribes Bactridinae and Attaleinae was confirmed. Also, the subtribe Elaeidinae appears more closely related to Bactridinae than to Attaleinae. In Bactridinae, *Bactris* and *Astrocaryum* are closely related genera, and in Attaleinae, *Cocos nucifera* as sister to *Syagrus coronata*.

## Discussion

### *Bactris gasipaes* plastome

The plastome of *Bactris gasipaes* has the typical quadripartite structure and gene content of other described Cocoseae species [5, 36–40]. The IRs are identical, which probably occurs due to mechanisms of replication and repair through recombination-dependent replication (RDR), as previously reported in plant plastomes [41]. The GC content in the plastomes of *Bactris gasipaes* and other species of Cocoseae corroborate the mean value described by Kwon et al. [42] for angiosperms (37.71; SD 1.10). The tendency for a higher GC content in the IRs than in the LSC and SSC was previously reported in bryophytes, ferns, lycophytes, and angiosperms [42, 43].

The gene content is conserved among species of Bactridinae (e.g., *Acrocomia aculeata*, *Astrocaryum murumuru*, *Astrocaryum aculeatum*, and *Bactris gasipaes*) and Elaeidinae (*Elaeis guineensis*) [36, 38, 39]. Attaleinae species present one pseudogenized *rps19*, and thus, one less CDS [40].

The gene *cemA* has an unconventional start codon in *Bactris gasipaes*, what was previously described in species of subtribes Attaleinae and Elaeidinae, in *Podococcus barteri* (NC_027276.1), *Phoenix dactylifera* 'Khalas', and other monocots. However, it is still not clear if this gene, with the unconventional start codon, is translatable to a protein [44]. Although most of the genes encoding proteins have ATG initiation codons [7], some alternative initiation codons are found in plants [45], such as GTG in the *rps19* gene, ACG in *rpl2* and ATC in *ndhD*, which were reported in *Lilium longiflorum*, *Phoenix dactylifera* 'Khalas' and *Amomum compactum*, respectively [44, 46, 47].

### Comparative plastome analysis within Cocoseae

Comparative studies using plastomes of species at different taxonomic levels can bring insights into plastome evolution, phylogenetic relationships, and evolutionary rates [2]. Plastomes of the three subtribes analyzed (Attaleinae, Bactridinae, and Elaeidinae), represented here by eight species, provided information to compare sequence variations in tribe Cocoseae. We identified slight differences in plastome size (~2 kb) and an inversion of 4.5 kp that occurs in the *ndh* complex (LSC region) of the *Astrocaryum* plastome. Similarly, Barrett et al. [48] reported that Arecaceae plastomes are highly conserved structure, describing only one 1.9 kb inversion located between the *rps16* and *trnG-UUC* genes in *Tahina spectabilis*.

Also, the variability among Attaleinae, Bactridinae, and Elaeidinae in the LSC/IR junctions, mainly involving the *rps19* gene, was previously described in *Acrocomia aculeata* [38] and *Butia eriosphata* [40], as well as in *Phoenix dactylifera* [44, 49]. Similarly, the *ndhF* gene overlapping with *ycf1* in ~25 bp is commonly observed in palms [38, 40, 44, 49]. Despite the differences in the IR junction, the IR structure and gene content is conserved among palms, corroborating the hypothesis that the IR regions offer an isolation mechanism that stabilizes the structure of the genome [50].

## Hypervariable regions

Plastomes have several non-coding regions, but not all of them have been explored for phylogenetic studies [3, 4, 28]. Among the ten regions with the greatest SV% identified in our study, only five IGSs (e.g., *accD-psaI*, *ndhF-rpl32*, *trnS-trnG*, *psaC-ndhE*, *rps15-ycf1*) and one intron (*rpl16*) were previously used and/or highlighted in angiosperm studies [3, 38, 51]. Among the ten regions with the greatest PIS%, only three IGS (*trnS-trnG*, *petA-psbJ*, and *psaC-ndhE*) were identified in studies carried out by Shaw et al. [3, 27] and Lopes et al. [38]. Thus, in our study, we described four new promising regions based on both SV and PIS values (*trnC-petN*, *psbC-trnS*, *ccsA-ndhD*, *petN-psbM*) and three new regions based on PIS (*petD-rpoA*, *trnG-trnfM*, *rps8-rpl14*). Also, Scarcelli et al. [51] reported 100 primers for phylogeny in monocots. However, four of the hypervariable regions reported in our study were not contemplated (e.g., *psaC-ndhE*, *petN-psbM*, *accD-psaI*, *trnS-trnG*). Among them. *trnS-trnG* and *accD-psaI* primers designed by Scarcelli et al. [51] showed no amplification in Arecaceae, and *petN-psbM* and *psaC-ndhE* were not mentioned, probably due to gene rearrangements in monocots plastomes. This reinforces that the highly variable regions vary between clades, and their identification may be necessary for distinct taxonomic levels. As expected, the nuclear genes PRK and RPB2 showed greater variation than most plastidial regions. These nuclear markers are very informative and produce well-resolved topologies [33]. The combined use of the plastidial regions described here and the nuclear markers PRK and RPB2 have great potential for phylogenetic studies in tribe Cocoseae.

## Phylogenomic analysis

The ten regions with the greatest SV% values are suitable for phylogenetic inferences and produce phylogenetic trees with well-resolved and the expected topologies. In the ML analysis, all datasets tested (plastome, ten SV regions, and ten PIS regions) result in subtribe Bactridinae as monophyletic. The monophyly Bactridinae was previouly described by Eiserhardt et al. [32], as well as the sister relationship between the subtribes Elaeidinae and Bactridinae [33, 38]. Our results are in contrast with those of Gunn [52], in which the sister relationship between *Astrocaryum* and *Bactris* is weakly supported. In all of our datasets this sister relationship is well-supported. In addition, the monophyly of subtribe Cocoseae was also verified in a plastid DNA analysis [53], in the super-tree method [54] and by the combined analysis with the PRK and RPB2 genes [33]. The sister relationship between *Cocos nucifera* and *Syagrus coronata* was also previously described [55], corroborating our results. Thus, both the plastome and the ten regions with greatest SV values were able to produce well-resolved phylogenetic trees and with consistent topologies within tribe Cocoseae.

## Supporting information

**S1 Table. GenBank accession numbers of the nucleotide sequences used in our analysis.** (DOC)

**S1 Fig. Phylogenetic tree based on maximum likelihood inference using the ten plastome regions with greatest PIS% values.** The numbers above the branches are maximum likelihood bootstrap values (1000 replicates). (TIF)

## Acknowledgments

We thank Jeison W. de S. Magnabosco and Michelle E. Zavala-Páez for their help with the analyses.

## Author Contributions

**Conceptualization:** Charles Roland Clement, Emanuel Maltempi de Souza, Hugo Pacheco de Freitas Fraga, Leila do Nascimento Vieira.

**Data curation:** Hugo Pacheco de Freitas Fraga, Leila do Nascimento Vieira.

**Formal analysis:** Raquel Santos da Silva, Eduardo Balsanelli, Valter Antonio de Baura, Hugo Pacheco de Freitas Fraga, Leila do Nascimento Vieira.

**Funding acquisition:** Charles Roland Clement, Emanuel Maltempi de Souza, Leila do Nascimento Vieira.

**Investigation:** Raquel Santos da Silva, Charles Roland Clement, Emanuel Maltempi de Souza, Leila do Nascimento Vieira.

**Methodology:** Raquel Santos da Silva, Eduardo Balsanelli, Valter Antonio de Baura, Hugo Pacheco de Freitas Fraga, Leila do Nascimento Vieira.

**Project administration:** Charles Roland Clement, Leila do Nascimento Vieira.

**Resources:** Charles Roland Clement, Eduardo Balsanelli, Leila do Nascimento Vieira.

**Software:** Valter Antonio de Baura.

**Supervision:** Charles Roland Clement, Emanuel Maltempi de Souza, Hugo Pacheco de Freitas Fraga, Leila do Nascimento Vieira.

**Validation:** Charles Roland Clement, Emanuel Maltempi de Souza, Hugo Pacheco de Freitas Fraga, Leila do Nascimento Vieira.

**Visualization:** Raquel Santos da Silva, Eduardo Balsanelli, Valter Antonio de Baura.

**Writing – original draft:** Raquel Santos da Silva.

**Writing – review & editing:** Raquel Santos da Silva, Charles Roland Clement, Hugo Pacheco de Freitas Fraga, Leila do Nascimento Vieira.

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
