## [Decision Letter · Decision Letter 0]

17 Jun 2021

PONE-D-21-11791

The plastome sequence of Bactris gasipaes and evolutionary analysis in tribe Cocoseae (Arecaceae)

PLOS ONE

Dear Dr. do Nascimento Vieira,

Thank you for submitting your manuscript to PLOS ONE. After careful consideration, we feel that it has merit but does not fully meet PLOS ONE’s publication criteria as it currently stands. Therefore, we invite you to submit a revised version of the manuscript that addresses the points raised during the review process.

Considering the reports of three reviewers, and my own assessment, I think a minor revision, which nevertheless answers to the suggestions in the reviews carefully, is the appropriate thing to do. Please check the details further below!

We look forward to receiving your revised manuscript.

Kind regards,

Berthold Heinze

Academic Editor

PLOS ONE

Journal Requirements:

1. Please ensure that your manuscript meets PLOS ONE's style requirements, including those for file naming. The PLOS ONE style templates can be found athttps://journals.plos.org/plosone/s/file?id=wjVg/PLOSOne_formatting_sample_main_body.pdf and https://journals.plos.org/plosone/s/file?id=ba62/PLOSOne_formatting_sample_title_authors_affiliations.pdf

Additional Editor Comments (if provided):

We have received three reviews for this manuscript. All authors emphasize that the study is straightforward and mostly well done. Normally, I am reluctant to recommend accepting manuscript which describes chloroplast sequencing, because this has become routine work, really (but be aware to rely too heavily on refernce-guided assembly! Have a look at a book chapter where I rather recommend to publish such data as "data reports": https://doi.org/10.1007/978-1-0716-0997-2_5).

So my recommendation is to include a Phoenix species, as some reviewers have done, and mention and discuss the study recommended (with the 100 primers for monocotyledon chloroplasts).

Reviewers' comments:

Reviewer's Responses to Questions

**Comments to the Author**

1. Is the manuscript technically sound, and do the data support the conclusions?

Reviewer #1: Yes

Reviewer #2: Yes

Reviewer #3: Yes

2. Has the statistical analysis been performed appropriately and rigorously? 

Reviewer #1: Yes

Reviewer #2: N/A

Reviewer #3: Yes

3. Have the authors made all data underlying the findings in their manuscript fully available?

Reviewer #1: Yes

Reviewer #2: No

Reviewer #3: Yes

4. Is the manuscript presented in an intelligible fashion and written in standard English?

Reviewer #1: Yes

Reviewer #2: Yes

Reviewer #3: Yes

5. Review Comments to the Author

Reviewer #1: This is a straightforward but sound description of the newly sequenced plastome of the economically important palm, Bactris gasipaes. The properties of the plastome are analysed in a simple phylogenetic framework including seven other species from tribe Cocoseae plus a small outgroup. The analysis is fine, the manuscript is well written and the discussion makes sense. My only comment is that the authors have overlooked a paper on phylogenetically informative plastid regions in palms: https://journals.plos.org/plosone/article?id=10.1371/journal.pone.0019954 I think the discussion would benefit from a comparison of the variable regions identified here to those of Scarcelli et al.

Reviewer #2: In this paper, authors sequenced and characterized the plastome of Bactris gasipaes (Bactridinae) and compared it with eight species from the three Cocoseae sub-tribes to perform comparative analysis and to identify hypervariable regions.

Reviewer #3: The main study objective is the use of the plastidial regions and the nuclear markers PRK and RPB2 for a suitable approach for phylogenetic inferences and produces phylogenetic trees with well-resolved and the expected topologies in tribe Cocoseae. Without using another subfamily far from the Cocoseae subtribe, e.g. P dactylifera, the conclusion cannot be drawn with confidence.

6. PLOS authors have the option to publish the peer review history of their article (what does this mean?). If published, this will include your full peer review and any attached files.

Reviewer #1: No

Reviewer #2: **Yes: **M. Kamran Azim

Reviewer #3: **Yes: **Ibrahim Al-Mssallem

---

## [Author Response · Author response to Decision Letter 0]

31 Jul 2021

Curitiba, July 31, 2021

Dear Dr. Berthold Heinze

Academic Editor of PLOS ONE

Manuscript ID: PONE-D-21-11791

Manuscript resubmission Requirement

We received the results of evaluation of the refereed MS. We thank the corrections and recommendations, which improved the MS overall quality. All the queries, recommendations and suggestions were considered, as follow:

Academic editor

We have received three reviews for this manuscript. All authors emphasize that the study is straightforward and mostly well done. Normally, I am reluctant to recommend accepting manuscript which describes chloroplast sequencing, because this has become routine work, really (but be aware to rely too heavily on reference-guided assembly! Have a look at a book chapter where I rather recommend to publish such data as "data reports": https://doi.org/10.1007/978-1-0716-0997-2_5). So my recommendation is to include a Phoenix species, as some reviewers have done, and mention and discuss the study recommended (with the 100 primers for monocotyledon chloroplasts).

R: Thank you for the positively evaluation and the constructive comments. We assembled the plastome using a de novo strategy. The reference-guided strategy was only used to order the contigs. The final assemble was verified by read mapping to ensure no contigs were mistakenly united. 

We have one species from subfamily Coryphoideae (the same subfamily of Phoenix) in the phylogenetic analysis. Also, Archontophoenix alexandrae (F.Muell.) H.Wendl. & Drude from subfamily Arecoideae, but Areceae subtribe.

We included the recommended study (Scarcelli et al. 2011) in the discussion. 

#Reviewer 1

This is a straightforward but sound description of the newly sequenced plastome of the economically important palm, Bactris gasipaes. The properties of the plastome are analysed in a simple phylogenetic framework including seven other species from tribe Cocoseae plus a small outgroup. The analysis is fine, the manuscript is well written and the discussion makes sense. My only comment is that the authors have overlooked a paper on phylogenetically informative plastid regions in palms: https://journals.plos.org/plosone/article?id=10.1371/journal.pone.0019954 I think the discussion would benefit from a comparison of the variable regions identified here to those of Scarcelli et al.

R: Thank you for the positively evaluation and the constructive comments. The discussion were improved and the following sentences were included: “Also, Scarcelli et al. [51] reported 100 primers for phylogeny in monocots. However, four of the hypervariable regions reported in our study were not contemplated (e.g., psaC-ndhE, petN-psbM, accD-psaI, trnS-trnG). Among them. trnS-trnG and accD-psaI primers designed by Scarcelli et al . [51] showed no amplification in Arecaceae, and petN-psbM and psaC-ndhE were not mentioned, probably due to gene rearrangements in monocots plastomes. This reinforces that the highly variable regions vary between clades, and their identification may be necessary for distinct taxonomic levels.”.

#Reviewer 2

In this paper, authors sequenced and characterized the plastome of Bactris gasipaes (Bactridinae) and compared it with eight species from the three Cocoseae sub-tribes to perform comparative analysis and to identify hypervariable regions.

R: Thank you for the positively evaluation.

#Reviewer 3

The main study objective is the use of the plastidial regions and the nuclear markers PRK and RPB2 for a suitable approach for phylogenetic inferences and produces phylogenetic trees with well-resolved and the expected topologies in tribe Cocoseae. Without using another subfamily far from the Cocoseae subtribe, e.g. P dactylifera, the conclusion cannot be drawn with confidence.

R: In the phylogenetic analysis, we included Brahea brandegeei (Purpus) H. E. Moore from the subfamily Coryphoideae (the same subfamily of Phoenix) and Archontophoenix alexandrae (F.Muell.) H.Wendl. & Drude from subfamily Arecoideae, but Areceae subtribe. These species as outgroup contemplates the suggestion. Also, we included the average depth of coverage of 67.64 (SD = 24.32).

Best regards,

Leila do Nascimento Vieira 

leilavieira@ufpr.br

Corresponding author

---

## [Editor Report · Decision Letter 1]

5 Aug 2021

The plastome sequence of Bactris gasipaes and evolutionary analysis in tribe Cocoseae (Arecaceae)

PONE-D-21-11791R1

Dear Dr. do Nascimento Vieira,

We’re pleased to inform you that your manuscript has been judged scientifically suitable for publication and will be formally accepted for publication once it meets all outstanding technical requirements.

Kind regards,

Berthold Heinze

Section Editor

PLOS ONE

Additional Editor Comments (optional):

Many thanks for submitting the revision. I see that all comments and recommendations were either followed up, or sufficiently discussed, in the letter and/or the manuscript itself. Also from a copyediting view, the manuscript seems ready for print.
---

## [Editor Report · Acceptance letter]

9 Aug 2021

PONE-D-21-11791R1 

The plastome sequence of Bactris *gasipaes* and evolutionary analysis in tribe Cocoseae (Arecaceae) 

Dear Dr. do Nascimento Vieira:

I'm pleased to inform you that your manuscript has been deemed suitable for publication in PLOS ONE. Congratulations! Your manuscript is now with our production department. 

Kind regards, 

on behalf of

Dr. Berthold Heinze 

Section Editor

PLOS ONE